# Optimization of Acute Kidney Injury (AKI) Time Definitions Using the Electronic Health Record: A First Step in Automating In-Hospital AKI Detection

**DOI:** 10.3390/jcm10153304

**Published:** 2021-07-27

**Authors:** Joshua T. Swan, Linda W. Moore, Harlan G. Sparrow, Adaani E. Frost, A. Osama Gaber, Wadi N. Suki

**Affiliations:** 1Department of Surgery, Houston Methodist Hospital, Houston, TX 77030, USA; LWMoore@houstonmethodist.org (L.W.M.); AOGaber@houstonmethodist.org (A.O.G.); 2Houston Methodist Academic Institute, Houston Methodist Hospital, Houston, TX 77030, USA; afrost@houstonmethodist.org (A.E.F.); wsuki@houstonmethodist.org (W.N.S.); 3Center for Outcomes Research, Houston Methodist Hospital, Houston, TX 77030, USA; 4Department of Pharmacy, Houston Methodist Hospital, Houston, TX 77030, USA; 5System Quality & Patient Safety, Houston Methodist Hospital, Houston, TX 77030, USA; hsparrow2@gmail.com; 6Department of Medicine, Houston Methodist Hospital, Houston, TX 77030, USA; 7Department of Medicine, Weill Cornell Medical College, New York, NY 77030, USA

**Keywords:** acute kidney injury, diagnosis, computer-assisted, epidemiology, detection limit, AKI timing

## Abstract

Kidney Disease: Improving Global Outcomes (KDIGO) acute kidney injury (AKI) definitions were evaluated for cases detected and their respective outcomes using expanded time windows to 168 h. AKI incidence and outcomes with expanded time intervals were identified in the electronic health records (EHRs) from 126,367 unique adult hospital admissions (2012–2014) and evaluated using multivariable logistic regression with bootstrap sampling. The incidence of AKI detected was 7.4% (*n* = 9357) using a 24-h time window for both serum creatinine (SCr) criterion 1a (≥0.30 mg/dL) and 1b (≥50%) increases from index SCr, with additional cases of AKI identified: 6963 from 24–48 h.; 2509 for criterion 1b from 48 h to 7 days; 3004 cases (expansion of criterion 1a and 1b from 48 to 168 h). Compared to patients without AKI, adjusted hospital days increased if AKI (criterion 1a and 1b) was observed using a 24-h observation window (5.5 days), 48-h expansion (3.4 days), 48-h to 7-day expansion (6.5 days), and 168-h expansion (3.9 days); all are *p* < 0.001. Similarly, the adjusted risk of in-hospital death increased if AKI was detected using a 24-h observation window (odds ratio (OR) = 16.9), 48-h expansion (OR = 5.5), 48-h to 7-day expansion (OR = 4.2), and 168-h expansion (OR = 1.6); all are *p* ≤ 0.01. Expanding the time windows for both AKI SCr criteria 1a and 1b standardizes and facilitates EHR AKI detection, while identifying additional clinically relevant cases of in-hospital AKI.

## 1. Introduction

In-hospital acute kidney injury (AKI) is associated with increased mortality, prolonged hospital stay, increased hospitalization costs, and adverse post-hospitalization outcomes [1,2,3,4,5]. Earlier reliable identification of AKI using the electronic health record (EHR) may allow prompt remedial intervention, tracking of hospital quality metrics, and clinical research initiatives. Early recognition of in-hospital AKI requires a consensus definition and ideally one that can be automated in the new electronic hospital environment. In order to provide a unified definition for detection and staging of AKI, the “Kidney Disease: Improving Global Outcomes” [6] (KDIGO) guidelines combined the recommendations of the Acute Dialysis Quality Initiative [7] in 2004 and the Acute Kidney Injury Network (AKIN) in 2007 [8]. The KDIGO guidelines define AKI as an increase in serum creatinine (SCr) by ≥0.3 mg/dL (≥26.5 umol/L) within 48 h (KDIGO 1a) or an increase in SCr to ≥1.5 times baseline within the prior 7 days (KDIGO 1b) [9] or urine volume < 0.5 mL/kg/h for 6 h [6]. As urine volume is poorly documented in the EHR during routine in-hospital care, the present investigation focuses exclusively on SCr criteria. Several guideline-making groups have critiqued the KDIGO AKI criteria and have requested additional research to evaluate these criteria [10,11,12,13]. The present study is intended to address two issues: first, the recognized arbitrariness of the choice of 48 h for the change of ≥0.3 mg/dL in creatinine (KDIGO 1a) since this change could conceivably occur earlier or later with different outcome implications; second, due to differences in the timing of blood draws between index SCr and follow up, 7 calendar days may not encompass the 168 h implied in KDIGO 1b for an increase of ≥50% from the index SCr. The primary objective of this study is to explore the impact (in terms of numbers of patients with AKI detected and their respective outcomes) of the ≥0.3 mg/dL increase in creatinine over 24-h intervals up to 168 h to identify cases of AKI that may occur earlier or later than the currently prescribed time. The secondary objective is to similarly explore the impact of changing the time frame for KDIGO 1b criteria for AKI and the ≥50% increase in SCr from index value to 168 h rather than 7 days. If this analysis reveals meaningful increases in the numbers of patients detected with AKI with associated meaningful differences in their outcomes, the case can be made for programming the EHR to automatically identify the thus defined AKI and to alert the health care team.

## 2. Materials and Methods

### 2.1. Study Design

This retrospective cohort study examined the impact of modifying time components of the KDIGO AKI criteria on the incidence of AKI and associations of AKI with clinical outcomes using existing EHR data from a large multi-hospital health system. The study was approved by the health-system’s institutional review board with a waiver of informed consent.

### 2.2. Patients and Setting

All patients admitted to a large academic medical center hospital and four satellite community hospitals within a health-system in Houston, Texas, from 1 January 2012 to 31 December 2014 were screened for inclusion. Exclusion criteria were as follows: age < 18 years; pregnancy; AKI defined as one of the following International Classification of Diseases Ninth Revision Clinical Modification (ICD-9-CM) codes that were present upon admission: 584.5, 584.6, 584.7, 584.8, or 584.9; CKD stage 5 present upon admission and identified using ICD-9-CM codes 585.5 or 585.6 or as an estimated glomerular filtration rate of <15 mL/min/1.73 m^2^ using the first SCr available during the hospitalization; fewer than two SCr assessments during the hospital stay; a history of nephrectomy or solid organ transplantation. Serum creatinine was measured with equipment calibrated using standards measured with isotope dilution mass spectrometry (IDMS).

### 2.3. Demographics and Baseline Comorbidities

Age, sex, and race/ethnicity were recorded. The presence of anemia, cancer, cerebrovascular disease, chronic kidney disease, cirrhosis, diabetes mellitus, hypertension, heart disease, liver disease, and other comorbidities was determined by ICD-9-CM coding upon admission. The Chronic Kidney Disease Epidemiology (CKD-Epi) equation [14] was used to estimate GRF (eGFR), which was reported in 15 mL/min/1.73 m^2^ increments [15].

### 2.4. Acute Kidney Injury

Acute kidney injury was detected using SCr only and defined as SCr criterion 1a (an absolute increase in the SCr of ≥0.3 mg/dL or ≥26.5 µmol/L) or SCr criterion 1b (≥50% relative increase). Observation windows were treated as rolling periods of time, hours, or days wherein all SCr assessments within each time period are compared for absolute and relative increases.

### 2.5. Identification of New Cases of AKI by Expanding Time Windows

This study undertook a stepwise expansion of observation windows for SCr criteria 1a and 1b up to 168 h (Appendix A). First, the analysis sought all cases of AKI that could be detected using a 24-h observation window for SCr criteria 1a and 1b. Second, additional cases of AKI were sought by expanding observation windows for SCr criteria 1a and 1b from 24 to 48 h (48-h expansion). Third, additional cases of AKI were sought by expanding the observation window for SCr criterion 1b only from 48 h to 7 days (48-h to 7-day expansion). Lastly, additional cases of AKI were sought by expanding the observation window for both SCr criteria 1a and 1b from 48 h to 168 h.

### 2.6. Observation Windows and Precision Levels

The study further evaluated 7 observation windows (1–7 days). For each observation window, three precision levels were evaluated (calendar day, 24-h period, and 24-h period plus a 4-h margin). The 4-h margin was added to account for variation in time of SCr measurements between days (Appendix A).

### 2.7. Narrowest Observation Window for Detecting AKI

To evaluate the stepwise expansion of observation windows and by using various precision levels, the “narrowest observation window for detecting AKI” was constructed and calculated using the time units of hours. For example, if two SCr assessments spaced 27 h apart increased from 1.0 to 1.4 mg/dL, criteria would not be met over a 24-h observation window and AKI would be detected using the “28 h” narrowest observation window.

### 2.8. Outcomes of Length of Stay and Mortality

Only in-hospital deaths were collected. Three hospital length of stay (LOS) metrics were calculated: (1) total LOS, (2) days before AKI, and (3) days after AKI.

### 2.9. Statistical Analysis

Logistic regression and linear regression were used to evaluate relationships between AKI and in-hospital mortality and LOS metrics, respectively. The associations of AKI expansion groups with outcomes were evaluated using adjusted regression with bootstrap sampling (1000 replications, with a sample size equal to the sample included in the adjusted regression) for internal validation [16]. Regression models were adjusted for hospital admission status (elective/unknown, urgent, and emergency), eGFR in increments of 15 mL/min/1.73 m^2^ based on first hospital SCr, gender, hospital type (academic medical center vs. community hospital), and baseline comorbidities. Models were not adjusted for baseline CKD or age since in the former eGFR was already included in the model and, in the latter, collinearity was detected between age and eGFR. Meaningful collinearity was not detected between gender and eGFR. A two-sided alpha of 0.05 was used to test for statistical significance. Analyses were performed using Stata version 16.1 (StataCorp LP, College Station, TX, USA). GraphPad Prism version 8.4 (GraphPad Software Inc., San Diego, CA, USA) was used to prepare study graphics.

Definitions of statistical terms are as follows: OR = odds ratio; 95% CI–95% confidence intervals; IQR = interquartile range.

## 3. Results

### 3.1. Patients and Setting

Of the 273,928 hospital admissions occurring during the study period, 126,367 admissions representing 83,938 unique patients were included in the study cohort (Figure 1 and Table 1) and 47% originated from the academic medical center. A diagnosis of CKD present upon admission was recorded for 9%, however, 24% had a baseline eGFR <60 mL/min/1.73 m^2^.

### 3.2. Identification of New Cases of AKI by Expanding Time Windows

The incidence of AKI detected was 7.4% (*n* = 9357) using a 24-h observation window for SCr criteria 1a and 1b. Additional cases of AKI were detected during the “48-h expansion” from 24 to 48 h (5.5%, *n* = 6963). A further 2509 cases of AKI (2.0%) were identified by expanding criteria 1b from 48 h to 7 days (48-h to 7-day expansion); and an additional 3004 cases were identified by expanding SCr criteria 1a and 1b from 48 to 168 h (the “168-h expansion”) (2.4%). The incidence of AKI detected therefore increased from 14.9% (*n* = 18,829) by using the standard AKI KDIGO criteria to 17.3% (*n* = 21,833) by using a 168-h observation window and thus captured a 16% (*n* = 3004) relative increase in “additional cases” of AKI.

### 3.3. Outcomes Associated with Time Windows: LOS and Mortality

Average hospital LOS was 4.7 ± 4.0 days among admissions where AKI was not detected. Average hospital LOS if AKI was detected using a 24-h observation window was 10.8 ± 10.4 days, 48-h expansion was 8.6 ± 7.3 days, 48-h to 7-day expansion was 11.8 ± 8.3 days, and 168-h expansion was 9.1 ± 6.0 days. Compared to patients without AKI, adjusted hospital LOS increases in those with AKI were: using a 24-h observation window, +5.5 days; (95% CI 5.3 to 5.7); using the 48-h expansion, +3.4 days (95% CI 3.2 to 3.6); using the 48-h to 7-day expansion, +6.5 days (95% CI 6.2 to 6.8); and using the 168-h expansion, +3.9 days (95% CI 3.7 to 4.1); all *p* < 0.001, (Figure 2).

Incidence of in-hospital death was 1.4% (1801 of 126,367) for all admissions and 0.6% (574 of 104,534) for admissions where AKI was not detected. Incidence of in-hospital death if AKI was detected using a 24-h time window was 9.6% (900 of 9357), the incidence for the 48-h expansion window was 3.4% (234 of 6963), the incidence using the 48-h to 7-day expansion was 2.4% (61 of 2509), and the incidence for the 168-hhour expansion from 48 h to 168 h for both criteria 1a and 1b was 1.1% (32 of 3004). Compared to patients without AKI, the adjusted risk of in-hospital death increased if AKI was detected using a 24-h window (odds ratio [OR] = 16.9), 48-h expansion (OR = 5.5), 48-h to 7-day expansion (OR = 4.2), and the 168-h expansion (OR = 1.6); all are *p* ≤ 0.01, (Figure 3).

Adjusted mean difference in days of hospital care (dots) and 95% confidence intervals (lines) were calculated using bootstrap sampled multivariable logistic regression, where the reference category were patients who did not develop AKI. Logistic regression was adjusted for hospital admission status (elective/unknown, urgent, and emergency), estimated GFR in categories of 15 mL/min/1.73 m^2^ based on first hospital serum creatinine, gender, hospital type (academic medical center vs. community hospital), and baseline comorbidities of diabetes, anemia, hypertension, congestive heart failure, liver disease, and peripheral and visceral atherosclerosis.

The cases of AKI detected using the 48-h to 7-day expansion or the 168-h expansion occurred later during the hospital stay compared to cases detected using the 24-h or 48-h observation windows (Figure 4). Therefore, a subgroup analysis was conducted among 34,245 admissions with a length of stay ≥7 days to minimize the influence of when the AKI was detected on associations with in-hospital mortality. In this subgroup, adjusted risk of in-hospital death increased if AKI was detected using a 24-h observation window (OR = 13.6), 48-h expansion window (OR = 5.7), 48-h to 7-day expansion window (OR = 3.3), and 168-h expansion window (OR = 1.8) compared to patients without AKI (all *p* ≤ 0.01).

### 3.4. SCr Measurement during Routine Care

A total of 673,011 SCr assessments were collected during the 126,367 included admissions and 68% (*n* = 460,154) of SCr assessments were collected between the hours of 3:00 a.m. and 7:59 a.m. (Appendix A). Of the 673,011 SCr assessments, 126,367 were the first (index) SCr recorded during the hospital stay and 546,644 were follow-up (non-index) assessments. The duration of time from the previous SCr assessment was 21 to 27 h for 53% (290,446 of 546,644) of non-index SCr assessments, representing fluctuations in the time of day that blood was collected each morning (Figure 5). Only 298,989 (54.7%) of non-index SCr assessments were collected ≤24 h from the previous SCr. Adding a 4-h margin to AKI calculations (from 24 to 28 h) captures an additional 150,523 (27.5%) of SCr assessments. This 4-h margin accounts for variability in clinical practice regarding the time of day that blood was collected on subsequent days and the impact of expanding observation windows by this 4-h margin was further investigated.

### 3.5. Observation Windows and Precision Levels

The incidence of AKI was greatly influenced by changes in observation windows and precision levels (Table 2). Each version of AKI identified meaningful associations with LOS and mortality and the major difference between versions was the incidence of AKI detected. The incidence of AKI increased from 1.9% to 17.3% as observation windows expanded from 1 calendar day to 172 h. Day-based precision systematically underestimated the incidence of AKI compared to hour-based precision. The addition of a 4-h margin detected a higher incidence of AKI. Compared to patients with no AKI, patients with AKI experienced an increase in hospital length by five or more days (all *p* < 0.001) regardless of the observation window or precision level used. Compared to patients with no AKI, patients who developed AKI that was detected by two SCr assessments on the same calendar day experienced a 20.1% absolute risk for in-hospital mortality (*p* < 0.001), although this rarely occurred (incidence of 1.9% of all admissions). The risk difference for mortality was 5.1% to 8.8% (all *p* < 0.001) for all other observation windows and precision levels.

### 3.6. Narrowest Observation Window for Detecting AKI and Associated Clinical Outcomes

Of the 21,894 cases of AKI detected using a 172-h observation window (largest window evaluated), 42.7% of cases were detected using a window of ≤24 h and 74.5% of AKI cases were detected using a window of ≤48 h (Figure 6). Compared to patients who did not develop AKI, total hospital LOS increased in a stepwise fashion for narrowest observation windows (all *p* < 0.001) (Appendix A). However, this trend was driven by increased days before AKI through the detection of additional cases of AKI that occurred later during the hospital course (Figure 7). The average hospital LOS following detection of AKI was ≥3.4 days for all narrowest observation windows.

Detection of AKI using a 24-h observation window was associated with tremendously increased risk for in-hospital death (adjusted OR = 17) (Appendix A and Figure 8). The detection of AKI using narrowest observation windows between 28 and 120 h was associated with increased risk for death (adjusted OR from 1.9 to 6.5).

### 3.7. Sensitivity Analysis of Impact of Death as a Competing Event for Hospital Length of Stay

Among survivors, death could not be a competing event and similar associations were observed for total hospital LOS, days before AKI detection, and days following AKI detection (Appendix A).

### 3.8. Sensitivity Analysis of the SCr Criteria Used for Detection of AKI

In the main analysis, both SCr criterion 1a (absolute increase in SCr of ≥0.3 mg/dL) and SCr criterion 1b (≥50% relative increase in SCr) were used to detect AKI at each narrowest observation window. Compared to the main analysis, similar associations were identified for hospital LOS and in-hospital death for both SCr criterion 1a and for SCr criterion 1b (Appendix A).

## 4. Discussion

Ideally, detection of in-hospital AKI would be prospective, automated, based on routinely collected hospital data, and available to the entire healthcare team. In order to advance clinical detection and research, the medical community requires a carefully crafted consensus definition of AKI using SCr criteria and instructions on how to reliably operationalize these SCr criteria to build clinical alerts, analyze clinical databases, or develop treatments or prevention strategies for AKI [17,18]. As a step towards this ideal, we previously reported the impact of magnitude of SCr change and suggested separating KDIGO AKI stage 1 into stage 1a and 1b [9]. The current study characterizes the time components of KDIGO AKI SCr criteria and explores how to operationalize these time components in a large clinical database.

This study demonstrated that expanding the timing windows of the KDIGO AKI criteria increases the incidence of AKI detected and is meaningfully associated with the outcomes of in-hospital death and LOS. An observation unique to the present study was the demonstration that increases in SCr over a 24-h period and particularly within one calendar day are associated with a substantial increased risk for mortality and increased hospital LOS. However only 54.7% of non-index SCr assessments were drawn less than 24 h from the previous SCr assessment, suggesting that AKI associated with very high mortality might have been more frequent than the reported 7.4% had there been more SCr levels drawn within 24 h.

### 4.1. Impact of Observation Windows

Expanding the observation windows beyond the traditional KDIGO criteria increased the incidence of AKI detected and, as with the traditional KDIGO criteria, these cases were still meaningfully associated with increased rates of in-hospital death and prolonged LOS. Extending SCr criterion 1a from the currently accepted 48-h window up to 168 h detected additional cases of AKI that were associated with increased risk for in-hospital death and prolonged LOS (Appendix A), providing evidence to expand the observation window for criterion 1a beyond 48 h. Compared to SCr criterion 1a, detection of AKI using SCr criterion 1b had larger associations with death and prolonged hospital stay following AKI, which is consistent with our previous findings [9]. When all of the study data are considered together, this study suggests that expanding the time interval for the diagnosis of in-hospital AKI KDIGO criterion 1a from 48 h to 168 h (with a matching 168-h observation period for criterion 1b) should be considered.

### 4.2. Impact of Precision Levels

Compared to using hour-level precision, the use of calendar days systematically underreports AKI. For example, the incidence of AKI detected using a three calendar day window (14.2%) is closer to the 52-h window (13.9%) rather than a 72-h window (15.1%). Therefore, if calendar days are used, the data advocates rounding up to one calendar day longer than the intended observation time. If narrow observation windows are chosen (e.g., 24 or 48 h), the 4-h margin allows for additional pairs of SCr to be compared, which improves surveillance and increases AKI detection. Of note, new cases of AKI that were detected using the 4-h margin occurred 1–2 days later than cases detected without the 4-h margin (Appendix A and Figure 7). If expanded observation windows are chosen (e.g., 168 h), the usefulness of the 4-h margin diminishes.

### 4.3. Strengths and Limitations

The strength of this study is its use of real-world clinical data from the EHR of multiple community hospitals and an academic medical center. The study did not rely on administrative data sets for coding of in-hospital AKI since these are fraught with error [19]. The data from this study provide insight into routine SCr monitoring that can be used to optimize timing components of AKI criteria for the development of real-time alerts or analysis of large EHR-based datasets. The approach in the present study uses rolling windows of observation rather than reliance on a pre-hospital baseline SCr, which was not available in the present study and is infrequently routinely available. Another limitation of this study was the unavailability of urine volume data. The study relies solely on SCr and is limited by the lack of more accurate verified biomarkers of AKI.

## 5. Conclusions

Increasing the observation window allows for the comparison of a greater number of SCr assessments which increases the incidence of AKI detected in-hospital. Detection of AKI using a 24-h window may represent a severe form of AKI (even for an absolute increase in SCr by ≥0.3 mg/dL). Cases of AKI detected using the expanded windows (e.g., >48 h) identified AKI later during the hospital course and continued to predict prolonged hospital stay and risk for in-hospital death. The importance of this data in clinical practice will be with the relatively simple programming of these AKI definitions into the electronic health record to generate clinical alerts. Electronic health record automated in-hospital expansion of the KDIGO AKI SCr observation time windows to 168 h increases the observed incidence of AKI from 14.9% to 17.3% and these additional patients with AKI experienced a clinically meaningful increased risk of in-hospital death and prolonged LOS.

## Figures and Tables

**Figure 1 jcm-10-03304-f001:**
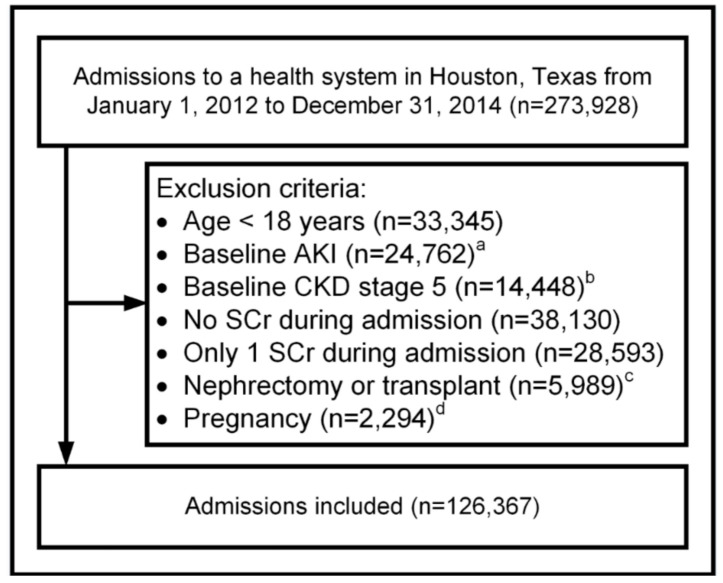
**Patient inclusion flowchart.** AKI, acute kidney injury; CKD, chronic kidney disease; SCr, serum creatinine. ^a^ AKI was defined as one of the following International Classification of Diseases Ninth Revision Clinical Modification (ICD-9-CM) codes that were present upon admission: 584.5, 584.6, 584.7, 584.8, or 584.9; ^b^ CKD stage 5 present upon admission was identified using ICD-9-CM codes (585.5 or 585.6) or as an estimated glomerular filtration rate of <15 mL/min/1.73 m^2^ using the first SCr available during the hospital stay; ^c^ Detected using principal procedures and the Agency for Healthcare Research and Quality (AHRQ) Clinical Classification Software (CCS) Procedure Class of 104 for nephrectomy, 105 for kidney transplant, and 176 for other types of solid organ transplant; ^d^ Detected using ICD-9-CM codes and the AHRQ clinical classification system categories 180 to 196.

**Figure 2 jcm-10-03304-f002:**
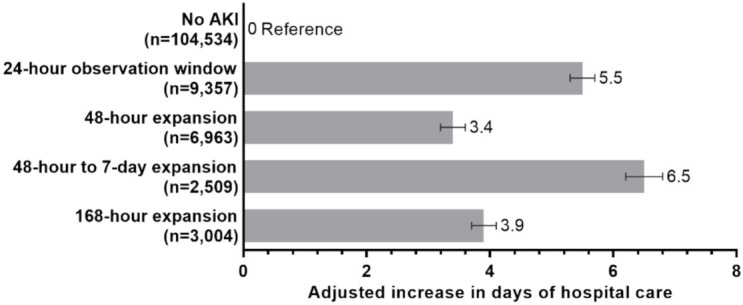
**Increased days of hospital care by observation window expansion (*n* = 126,367).** AKI, acute kidney injury; KDIGO: Kidney Disease Improving Global Outcomes. Adjusted mean difference in days of hospital care and 95% confidence intervals (lines) were calculated using bootstrap sampled multivariable logistic regression, where the reference category was patients who did not develop AKI. Logistic regression was adjusted for hospital admission status (elective/unknown, urgent, and emergency), estimated GFR in categories of 15 mL/min/1.73 m^2^ based on first hospital serum creatinine, gender, hospital type (academic medical center vs. community hospital), and baseline comorbidities of diabetes, anemia, hypertension, congestive heart failure, liver disease, and peripheral and visceral atherosclerosis.

**Figure 3 jcm-10-03304-f003:**
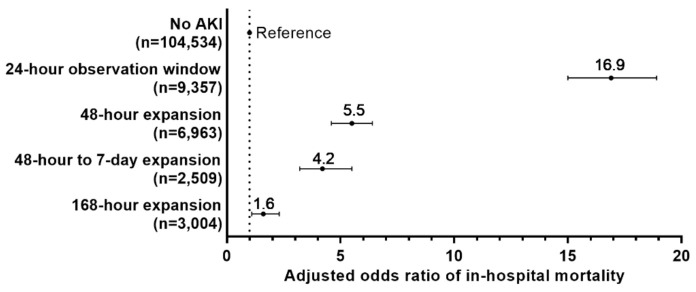
**Increased odds of in-hospital mortality by observation window expansion (*n* = 126,367).** AKI, acute kidney injury; KDIGO, Kidney Disease: Improving Global Outcomes. Adjusted odds ratios (dots) and 95% confidence intervals (lines) were calculated using bootstrap sampled multivariable logistic regression where the reference category were patients who did not develop AKI. Logistic regression was adjusted for hospital admission status (elective/unknown, urgent, and emergency), estimated GFR in categories of 15 mL/min/1.73 m^2^ based on first hospital serum creatinine, gender, hospital type (academic medical center vs. community hospital), and baseline comorbidities of diabetes, anemia, hypertension, congestive heart failure, liver disease, and peripheral and visceral atherosclerosis.

**Figure 4 jcm-10-03304-f004:**
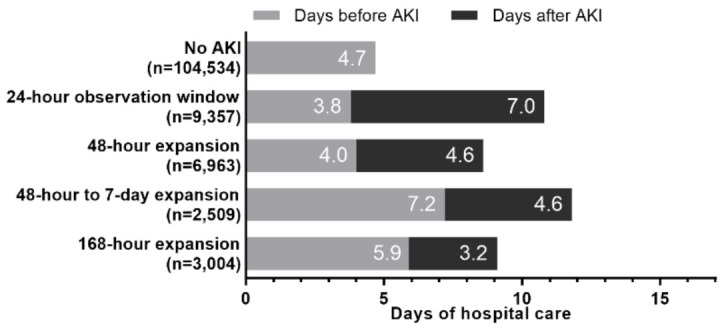
**Days of hospital care before and after the detection of AKI by observation window expansion (*n* = 126,367).** AKI, acute kidney injury; KDIGO, Kidney Disease: Improving Global Outcomes. The days before AKI were calculated as days from hospital admission to first detection of AKI. If AKI was not detected, the days before AKI were equal to the entire length of hospital stay. Days after AKI were the number of days from first detection of AKI to hospital discharge. Days after AKI does not exist for patients who did not develop AKI; therefore, adjusted regression could not be used, and unadjusted data are presented in this figure

**Figure 5 jcm-10-03304-f005:**
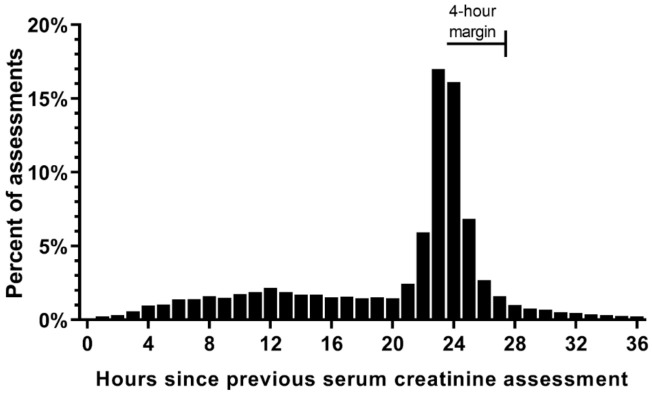
**Histogram of hours between two serial serum creatinine assessments**. The x-axis displays the number of hours since the last serum creatinine (SCr) assessment rounded down to the nearest hour. The bar at “0” represents a time of 1 to 59 min since the last SCr assessment. For example, if SCr was measured at 04:00 a.m. on hospital day 1 and 04:30 a.m. on hospital day 2, the delta time between those assessments would be 24.5 h, which would be displayed on the bar at “24” on this figure. Of the 546,644 non-index SCr assessments that had a calculated time since last assessment, 87.5% were collected less than 37 h since the previous assessment. Data for 68,274 (12.5%) assessments that were collected more than 37 h since the previous assessment are not shown on this figure.

**Figure 6 jcm-10-03304-f006:**
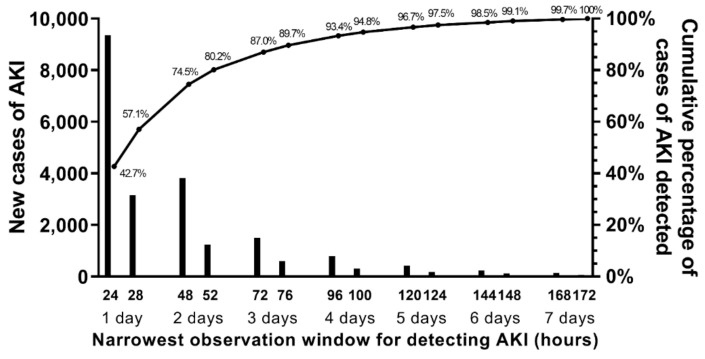
**New cases of AKI detected through stepwise expansion of observation window from 24 h to 172 h (*n* = 21,894).** AKI, acute kidney injury. The bars display the number of new cases of AKI that are detected as the rolling observation window (used to calculate the increase from lowest to highest serum creatinine) expanded from 24 h to 172 h. Count data are shown on the left y-axis. For example, 9357 cases of AKI that were detected using a rolling 24-h window and are shown in the first bar. The second bar at 28 h shows the 3146 new cases that were detected using a 28-h rolling window that were not previously detected using the 24-h rolling window. By moving right on the x-axis, each bar shows the number of new cases that were not previously detected by narrower observation windows. Cumulative percentage is displayed on the right y-axis. The solid line represents the cumulative percentage of the 21,894 cases of AKI detected at each specific observation window using data from that observation window and all narrower observation windows.

**Figure 7 jcm-10-03304-f007:**
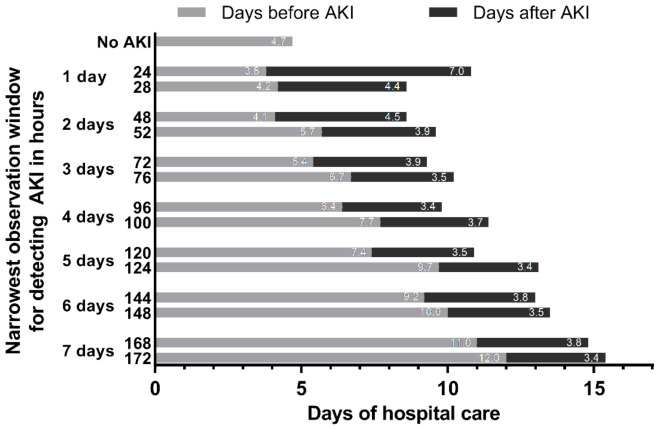
**Days of hospital care before and after detection of AKI by narrowest observation window for detecting AKI (*n* = 126,367).** AKI, acute kidney injury. Days before AKI was calculated as days from hospital admission to first detection of AKI. If AKI was not detected, days before AKI was equal to the entire length of hospital stay. Days after AKI was the number of days from first detection of AKI to hospital discharge. Days prior to hospital AKI do not exist for patients who did not develop AKI; therefore, adjusted regression could not be used, and unadjusted data are presented in this figure. Differences in total length of stay were driven by differences in days before AKI detection. As the observation window for detection of AKI increases from 24 h to 172 h, additional cases of AKI were detected later during hospital admission. The average hospital length of stay following the detection of AKI was 3.4 days or longer for all observation windows.

**Figure 8 jcm-10-03304-f008:**
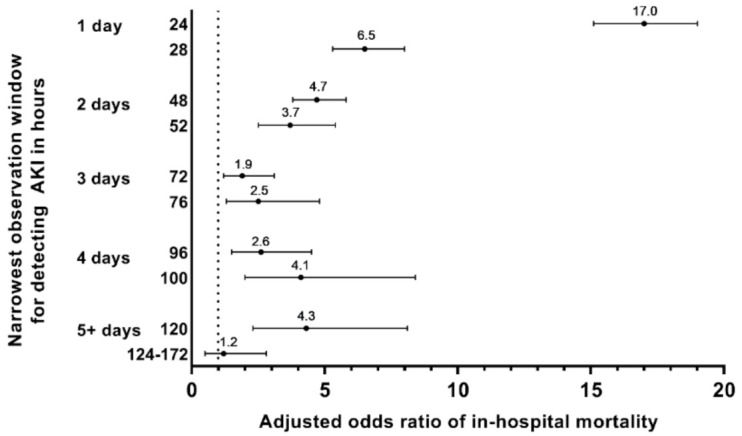
**Adjusted odds ratio of in-hospital mortality by narrowest observation window for detecting AKI (*n* = 126,367).** AKI, acute kidney injury. Adjusted odds ratios (dots) and 95% confidence intervals (lines) were calculated using logistic regression, where the reference category were patients who did not develop AKI. Logistic regression was adjusted for hospital admission status (elective/unknown, urgent, and emergency), estimated GFR in categories of 15 mL/min/1.73 m^2^ based on first hospital serum creatinine, gender, hospital type (academic medical center vs. community hospital), and baseline comorbidities of diabetes, anemia, hypertension, congestive heart failure, liver disease, and peripheral and visceral atherosclerosis.

**Table 1 jcm-10-03304-t001:** Demographic and clinical characteristics of admissions to a hospital system evaluated for the incidence of acute kidney injury.

Characteristic	Study Sample (*n* = 126,367)	
**Age, years**		
Mean (SD)	63	(18)
Median (range)	64	(18 to 112)
**Serum creatinine assessments**		
Count of SCr during admission, Median (IQR)	4	(3 to 6)
SCr per day of hospital care, mean (SD), mg/dL	1.1	(0.6)
	*n*	%
**Hospital type**		
Community hospital ^a^	67,175	53.2%
Academic medical center	59,192	46.8%
**Admission type**		
Elective	22,018	17.4%
Urgent	63,069	49.9%
Emergency	41,119	32.5%
Unavailable	144	0.1%
**Sex**		
Female	69,698	55.2%
Male	56,669	44.8%
**Race**		
Caucasian	77,373	61.2%
Black or African American	24,403	19.3%
Other	24,591	19.5%
**Estimated GFR using first SCr ^b^, mL/min/1.73 m^2^**		
15 to 29	3178	2.5%
30 to 44	9801	7.8%
45 to 59	16,762	13.3%
60 to 74	20,098	15.9%
75 to 89	24,857	19.7%
90 to 105	25,276	20.0%
>105	26,395	20.9%
**Chronic comorbid conditions ^c^**		
Hypertension	84,771	67.1%
Heart disease	59,863	47.4%
Congestive heart failure ^d^	23,317	18.5%
Diabetes	38,835	30.7%
Anemia	18,706	14.8%
Cancer	16,106	12.8%
Cerebrovascular disease	13,301	10.5%
Chronic kidney disease	11,820	9.4%
Other kidney/urinary tract disease	24,703	9.0%
Peripheral and visceral atherosclerosis	8167	6.5%
Liver disease	7759	6.1%
Pancreas disease (not diabetes)	1326	0.5%
**Admission sequence ^e^**		
First (index) admission for a patient	83,938	66.4%
Second admission for a patient	21,700	17.2%
Greater than second admission for a patient	20,729	16.4%

GFR, glomerular filtration rate; IQR, interquartile range; SCr, serum creatinine; SD, standard deviation. All information is displayed at the level of hospital admission and not at the level of a unique patient. The conversion factor for units of SCr in mg/dL to µmol/L is x88.4. ^a^ Admission to one of our community hospitals in the health system. ^b^ The 2009 Chronic Kidney Disease Epidemiology Collaboration (CKD-EPI) creatinine-based equation and the first in-hospital serum creatinine were used to estimate glomerular filtration rate. Patients with an estimated glomerular filtration rate of <15 mL/min/1.73 m^2^ were excluded from the study. ^c^ Chronic comorbid conditions were restricted to International Classification of Diseases Ninth Revision Clinical Modification codes that were flagged as chronic conditions using the Agency for Healthcare Research and Quality (AHRQ) Chronic Condition Indicator (CCI) version 2015. The AHRQ diagnosis clinical classification system (CCS) version 2015 categories 11–43 or 45 for cancer, 49–50 for diabetes, 59–61 for anemia, 96 or 100–108 for heart disease, 108 for congestive heart failure (subgroup of heart disease), 98–99 for hypertension, 109–113 for cerebrovascular disease, 114 for peripheral and visceral atherosclerosis, 6 and 150–151 for liver disease, 152 for pancreas disease (other than diabetes), 156 and 159–164 for other kidney disease, and 158 for chronic kidney disease. Comorbid conditions are not mutually exclusive. ^d^ Congestive heart failure is a subgroup of heart disease. ^e^ These data are presented by hospital admission and the admission sequence variable provides information on how many times a patient was included in the study cohort. The study included 83,938 unique patients representing a total of 126,367 unique hospital admissions.

**Table 2 jcm-10-03304-t002:** Incidence of acute kidney injury based on observation window and precision level with the outcomes length of stay and in-hospital mortality (*n* = 126,367).

Observation Window and Precision Level	AKI Incidence		Mean Length of Stay in Days ^a^				In-Hospital Mortality Incidence			
	*n*	(%)	AKI	No AKI	Unadjusted Difference in Means ^b^	95% CI	AKI	No AKI	Unadjusted Risk Difference ^c^	95% CI
**1 day**										
1 cal day	2381	1.9%	13.8	5.5	8.4 *	8.2 to 8.6	21.1%	1.0%	20.1% *	18.4% to 21.7%
24 h	9357	7.3%	10.8	5.2	5.6 *	5.5 to 5.7	9.6%	0.8%	8.8% *	8.2% to 9.4%
28 h ^d^	12,509	9.9%	10.2	5.1	5.1 *	5.0 to 5.2	8.2%	0.7%	7.5% *	7.0% to 8.0%
**2 days**										
2 cal days	13,271	10.5%	10.2	5.1	5.1 *	5.0 to 5.2	8.0%	0.7%	7.3% *	6.9% to 7.8%
48 h	16,320	12.9%	9.8	5.0	4.8 *	4.8 to 4.9	6.9%	0.6%	6.3% *	5.9% to 6.7%
52 h ^d^	17,559	13.9%	9.8	4.9	4.9 *	4.8 to 5.0	6.6%	0.6%	6.0% *	5.7% to 6.4%
**3 days**										
3 cal days	17,892	14.2%	9.8	4.9	4.9 *	4.8 to 5.0	6.5%	0.6%	5.9% *	5.6% to 6.3%
72 h	19,053	15.1%	9.8	4.9	4.9 *	4.8 to 5.0	6.2%	0.6%	5.6% *	5.3% to 6.0%
76 h ^d^	19,649	15.6%	9.8	4.8	4.9 *	4.9 to 5.0	6.1%	0.6%	5.5% *	5.2% to 5.8%
**4 days**										
4 cal days	19,831	15.7%	9.8	4.8	5.0 *	4.9 to 5.0	6.0%	0.6%	5.5% *	5.1% to 5.8%
96 h	20,440	16.2%	9.8	4.8	5.0 *	4.9 to 5.1	5.9%	0.6%	5.3% *	5.0% to 5.6%
100 h ^d^	20,749	16.4%	9.8	4.8	5.0 *	5.0 to 5.1	5.8%	0.6%	5.3% *	5.0% to 5.6%
**5 days**										
5 cal days	20,856	16.5%	9.8	4.8	5.0 *	5.0 to 5.1	5.8%	0.6%	5.3% *	5.0% to 5.6%
120 h	21,168	16.8%	9.8	4.8	5.1 *	5.0 to 5.2	5.8%	0.6%	5.2% *	4.9% to 5.5%
124 h ^d^	21,347	16.9%	9.9	4.7	5.1 *	5.0 to 5.2	5.7%	0.6%	5.2% *	4.9% to 5.5%
**6 days**										
6 cal days	21,396	16.9%	9.9	4.7	5.1 *	5.0 to 5.2	5.7%	0.5%	5.2% *	4.9% to 5.5%
144 h	21,573	17.1%	9.9	4.7	5.2 *	5.1 to 5.2	5.7%	0.6%	5.1% *	4.8% to 5.4%
148 h ^d^	21,960	17.2%	9.9	4.7	5.2 *	5.1 to 5.3	5.6%	0.6%	5.1% *	4.8% to 5.4%
**7 days**										
7 cal days	21,718	17.2%	9.9	4.7	5.2 *	5.1 to 5.3	5.6%	0.6%	5.1% *	4.8% to 5.4%
168 h	21,833	17.3%	10.0	4.7	5.2 *	5.2 to 5.3	5.6%	0.5%	5.1% *	4.8% to 5.4%
172 h ^d^	21,894	17.3%	10.0	4.7	5.3 *	5.2 to 5.3	5.6%	0.5%	5.1% *	4.7% to 5.4%
**KDIGO**										
7 days ^e^	18,829	14.9%	10.1	4.8	5.3 *	5.2 to 5.3	6.3%	0.6%	5.8% *	5.4% to 6.1%

cal, calendar; KDIGO, Kidney Disease Improving Global Outcomes; SCr, serum creatinine. At each observation window, AKI was detected if there was an absolute increase in SCr by ≥0.3 mg/dL or a relative increase in SCr by ≥50% over the specified rolling observation window set by the specified precision level. The reference group of patients who did not meet AKI criteria was unique for each precision level. * *p* value < 0.001. ^a^ Length of stay was calculated as the date and time of hospital discharge minus the date and time of hospital admission. ^b^ Calculated as mean length of stay among patients with AKI minus mean length of stay among patients who did not meet AKI. ^c^ The absolute risk difference was calculated as incidence of death among patients with AKI minus incidence of death among patients without AKI. ^d^ This precision level includes an extra 4-h margin to account for process variability in the time of SCr measurement during routine medication care. ^e^ The KDIGO criteria are applied as an absolute increase in SCr by ≥0.3 mg/dL over a 48-h rolling window or a relative increase in SCr by ≥50% over a 7-day rolling window. No urine output data are included in the study.

## Data Availability

The data presented in this study are available on request from the first author. The data are not publicly available due to privacy concerns.

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
