# Peer review of "Optimization of Acute Kidney Injury (AKI) Time Definitions Using the Electronic Health Record: A First Step in Automating In-Hospital AKI Detection"

_jcm, 2021, doi:10.3390/jcm10153304_

Round 1
Reviewer 1 Report
Topic is interesting and study well performed. Authors need to clearly report primary and secondary outcomes in order to improve efficacy of presentation. Moreover, it has to be stressed the power of results obtained in the clinical practice.
Author Response
Topic is interesting and study well performed.
The authors thank this reviewer for this comment.
Authors need to clearly report primary and secondary outcomes in order to improve efficacy of presentation.
Thank you. We have changed the wording in the introduction as follows:
The primary objective of this study is to explore the impact (in terms of numbers of patients detected and their respective outcomes) of the >0.3 mg/dl increase in creatinine over 24 hr intervals up to 168 hrs. to identify cases of AKI that may occur earlier or later than the currently proscribed time; and the secondary objective is to similarly explore the impact of changing the time frame for KDIGO 1b criteria for AKI, the >50% increase in SCr from index value to 168 hrs. rather than 7 days.
Moreover, it has to be stressed the power of results obtained in the clinical practice.
Thank you for this comment.
To emphasize the importance of the findings in clinical practice the following sentence was added to the Conclusion: "The importance of this data in clinical practice will be with the relatively simple programming of these AKI definitions into the electronic health record to generate clinical alerts."
Reviewer 2 Report
AKI, which is missed by the KDIGO AKI criteria due to time window issues, was also associated with prolonged hospital stay and death during hospitalization. The main message in this study that extending the time window would lead to improved diagnosis of AKI during hospitalization was accepted.
However, I found the amount of information (especially figures) too much to understand. I thought it would be better to reduce the amount of information if it could be focused only on important figures.
The following is a list of minor points, but I hope it will be helpful.
The abbreviations for EHR in the Key points and Abstract are not explained.
In the abstract and in the text, there is a description of the Odds ratio, but I think the 95% CI should also be described.
The reference in the text is a mixture of Roman numerals and Arabic numerals.
In the exclusion criteria, the authors forgot to separate <18 years and pregnancy with ;.
I think it would be easier to read if each item in the Table1 row was centered and left-aligned.
The font of the footnotes in Figures 2, 3, and 4 is somehow different and the text is too large, although this may be fixed in the publication stage.
In Figure 7, the difference between the footnote and the text is not clear.
Author Response
I found the amount of information (especially figures) too much to understand. I thought it would be better to reduce the amount of information if it could be focused only on important figures.
It appears that the reviewer received a copy of the manuscript formatted by the editorial staff which regrettable included the following errors:
switching from Arabic to roman numerals for end notes
changing the format of several tables resulting in crowding of the columns
interspersal of the text of the manuscript with randomly inserted figures and tables, and disruption of the normal headings ( eg Discussion). This is not the document which we submitted, nor the document we are working from to rectify any errors, or make modifications as per the editorial staff directive.
The following is a list of minor points, but I hope it will be helpful.
The abbreviations for EHR in the Key points and Abstract are not explained.
In the abstract and in the text, there is a description of the Odds ratio, but I think the 95% CI should also be described.
Thank you . We have added a sentence in Methods to define all statistical abbreviations. Definitions of statistical terms: OR = odds ratio; 95% CI – 95% confidence intervals; IQR = interquartile range
The reference in the text is a mixture of Roman numerals and Arabic numerals.
see first comment
In the exclusion criteria, the authors forgot to separate <18 years and pregnancy with ;.
Thank you we have added a comma.
I think it would be easier to read if each item in the Table1 row was centered and left-aligned.
Thank you but this was an editorial office formatting error of some sort and we can't address as it does not exist in the manuscript as submitted and resubmitted.
The font of the footnotes in Figures 2, 3, and 4 is somehow different and the text is too large, although this may be fixed in the publication stage.
In Figure 7, the difference between the footnote and the text is not clear.
Again these changes are editorial formatting changes that were not present in the manuscript as submitted which had conventional formatting for figure and table footnotes ( smaller font, and alphabetically identified)